# Non-identifiability and the Blessings of Misspecification in Models of Molecular Fitness

**Eli N. Weinstein**[*][†]
Columbia University
ew2760@columbia.edu

**Alan N. Amin**[*]
Harvard Medical School
alanamin@g.harvard.edu

**Jonathan Frazer**
Harvard Medical School and
Centre for Genomic Regulation (CRG)
jonathan.frazer@crg.eu

**Debora S. Marks**
Harvard Medical School and
Broad Institute of Harvard and MIT
debbie@hms.harvard.edu

## Abstract

Understanding the consequences of mutation for molecular fitness and function is a fundamental problem in biology. Recently, generative probabilistic models have emerged as a powerful tool for estimating fitness from evolutionary sequence data, with accuracy sufficient to predict both laboratory measurements of function and disease risk in humans, and to design novel functional proteins. Existing techniques rest on an assumed relationship between density estimation and fitness estimation, a relationship that we interrogate in this article. We prove that fitness is not identifiable from observational sequence data alone, placing fundamental limits on our ability to disentangle fitness landscapes from phylogenetic history. We show on real datasets that perfect density estimation in the limit of infinite data would, with high confidence, result in poor fitness estimation; current models perform accurate fitness estimation because of, not despite, misspecification. Our results challenge the conventional wisdom that bigger models trained on bigger datasets will inevitably lead to better fitness estimation, and suggest novel estimation strategies going forward.

## 1 Introduction

The past decades have witnessed a tremendous increase in the scale of genome sequence data available from across life. Recently, methods for estimating molecular fitness using generative sequence models have seen widespread success at translating this evolutionary data into predictions of the functional consequences of mutation. Such models have been shown to accurately predict the outcomes of experimental assays of protein function [23, 44, 36], and have been applied to infer 3D structures of RNA and protein [35, 56] and to design novel proteins [50, 47, 33]. The models have also been used to predict whether human mutations are pathogenic, directly informing the diagnosis of genetic disease [19]. In this paper, we investigate how and why generative sequence models fit to evolutionary sequence data are successful at estimating molecular fitness, and how they might be improved and generalized going forward.

Existing approaches to fitness estimation with generative sequence models rest on an assumed relationship between density estimation and fitness estimation. Given a dataset of sequences $X_1, \ldots, X_N$, assumed to be drawn i.i.d. from some underlying distribution $p_0$, fitness models proceed by (1)

---

[*]These authors contributed equally.

[†]Work done while at Harvard Medical School.

36th Conference on Neural Information Processing Systems (NeurIPS 2022).

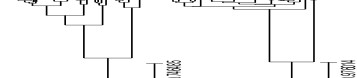

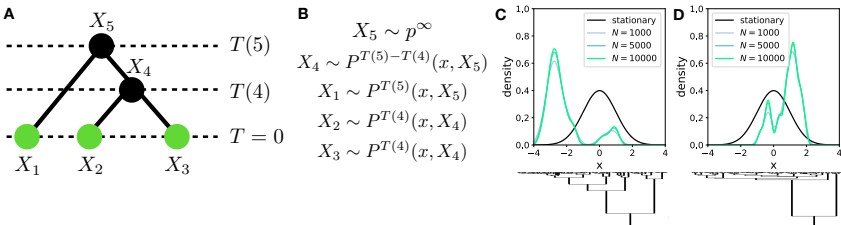

Figure 1: **JFPM illustration.** (A) Example JFPM for $N = 3$ observed sequences. (B) Generative process for sequences at each node of the phylogeny **H**. (C) *Above:* Stationary distribution $p^\infty$ and kernel density estimates of the distribution of samples $p_0$ from an OUT model for increasing $N$. *Below:* A subset of the phylogeny. (D) Same as (C) for an independent sample of **H**.

fitting a probabilistic model $q_\theta$ to $X_{1:N}$ and (2) using the inferred density $\log q_{\hat\theta}(x) \approx \log p_0(x)$ as an estimate of the fitness $f(x)$ of a sequence $x$; this estimate in turn is used to predict other covariates such as whether the mutated sequence is pathogenic [23, 44, 19]. Innovation in fitness models has come out of a trend of building increasingly flexible models fit to increasing amounts of data: simple models that treat each column of a sequence alignment independently were improved by energy-based models that accounted for epistasis [23], which in turn were improved by deep variational autoencoders [44], which in turn were improved by deep autoregressive alignment-free models [50, 33, 36]. Naively, one might assume that these improvements have come from obtaining better and better estimates of the data distribution $p_0$, and improvements will continue with bigger models and bigger datasets. In this article, we argue that this presumption is incorrect.

**Technical summary** First, we show that that the true data distribution $p_0$ may not reflect fitness, and argue instead that we should be focused on estimating another distribution that does, $p^\infty$ (the "stationary distribution", to be defined below). In particular, we demonstrate that phylogenetic effects – i.e. the history of how current sequences evolved over time – can "distort" the observed data, leading to a situation where $p_0 \neq p^\infty$ (Sec. 2). Second, we show in this situation that $p^\infty$ and fitness $f$ are non-identifiable: even with infinite data, there always exists some alternative fitness function $\tilde f$ that explains the same data just as well as $f$. This sets fundamental limits on what we can learn about fitness from evolutionary data (Sec. 3). Third, although exact estimation of $p^\infty$ is impossible, we show that it is still possible to get closer to $p^\infty$ than $p_0$, that is, to find a better estimator of fitness than the true data density $p_0$. This can be done by fitting to data a parametric generative sequence model $\mathcal{M} = \{q_\theta : \theta \in \Theta\}$ that is (approximately) well-specified with respect to $p^\infty$ (i.e. $p^\infty \in \mathcal{M}$) but *misspecified* with respect to the data distribution $p_0$ (i.e. $p_0 \notin \mathcal{M}$), thus illustrating how misspecification can be a blessing rather than a curse (Sec. 4). Fourth, we construct a hypothesis test to determine whether the blessings of misspecification occur on real data, for existing fitness estimation models; our test uses a recently developed Bayesian nonparametric sequence model to construct a credible set for $p_0$ (Sec. 6). Fifth, we apply our test to over 100 separate sequence datasets and fitness estimation tasks, to conclude that existing fitness estimation models systematically outperform the true data distribution $p_0$ at estimating fitness (Sec. 7). The takeaway is that better fitness estimation (i.e. better $p^\infty$ estimation) will not come from better density estimation (i.e. better $p_0$ estimation); bigger models and bigger datasets are not enough. Instead, better fitness estimation can come from developing models that describe $p^\infty$ better but the data density $p_0$ *worse*.

## 2   Models of Fitness and Phylogeny

In this section we illustrate how $p_0$ may not accurately reflect the true fitness landscape, by analyzing a generative model of sequence evolution that takes into account not only fitness but also phylogeny. Our model is general: it allows for arbitrarily complex epistatic fitness landscapes, and recovers standard phylogenetic and fitness models as special cases. Our concerns about the effects of phylogeny on fitness estimation are motivated by the widespread use – and trust – of phylogenetic models for evolutionary sequence data (phylogenetic models are far more widely applied than fitness models) [21, 12, 17, 18]. Note, however, that phylogeny is only one reason why $p_0$ may not not reflect the true fitness landscape, and that our later results on the benefits of misspecification in fitness models (Sec. 4) do not depend on the specific cause of mismatches between $p_0$ and the fitness landscape.

**Joint fitness and phylogeny models** We define "joint fitness and phylogeny models (JFPMs)" using two elements: a description of how individual species (or populations or individuals) change over time, which depends on fitness $f$, and a description of the species' relationship to one another, a phylogeny $\mathbf{H}$. To describe the dynamics of individual species, let $P^\tau(x, x_0)$ denote the probability of sequence $x_0$ evolving into sequence $x$ after time $\tau$; in particular, $P^\tau(x, x_0)$ is assumed to be the transition probability of an irreducible continuous-time Markov chain defined over sequence space $\mathcal{X}$. For example, under neutral evolution (i.e. without selection based on fitness), $P^\tau(x, x_0)$ may follow a Jukes-Cantor model [18]. With selection, for simple population genetics models (e.g. Moran or Wright processes), Sella and Hirsh [49] demonstrate under general conditions that for any $x_0$,

$$P^\tau(x, x_0) \xrightarrow{\tau \to \infty} p^\infty(x) = \frac{1}{\mathcal{Z}} \exp(\beta f(x)) \tag{1}$$

where $f(x)$ is the log fitness of the sequence $x$ and $\beta > 0$ is a constant (Appx. B). The implication of Eqn. 1 is that the stationary distribution of the evolutionary dynamics follows a Boltzmann distribution, with energy proportional to the log fitness of the sequence. Estimating $p^\infty$ is of interest because it provides a direct estimate of log fitness, up to a linear transform, since $f(x) = \beta^{-1}(\log p^\infty(x) + \log \mathcal{Z})$. (N.b. in the remainder of the paper, when we say "estimate fitness" we mean, implicitly, "estimate log fitness up to a linear transform".)

The sequences we observe, however, do not necessarily come from the stationary distribution. Instead, they are correlated with one another according to their evolutionary history. This is described by a phylogeny $\mathbf{H} = (V, E, T)$ consisting of a directed and rooted full binary tree with edges $E$ and nodes $V$, along with time labels for the nodes, $T : V \to \mathbb{R}_+$ (Fig. 1A). Each node $v$ is associated with a sequence $X_v$, drawn as $X_v \sim P^{\Delta t}(x, X_{v_0})$, where $X_{v_0}$ is the sequence of the parent node, $v$ is the child node, and $\Delta t = T(v_0) - T(v)$ is the length of the edge between them (Fig. 1B). The root sequence is drawn from $p^\infty$. The observed datapoints $X_1, \ldots, X_N$ correspond to the leaf nodes. In general we will assume all leaves are observed at effectively the same time, the present day $T = 0$.

**Special cases** Standard probabilistic phylogenetic models ignore fitness and assume

**Assumption 2.1** (Pure phylogeny models (PMs)). *Constant fitness:* $f(x) = C$.

Example models that fit this form include most of those used in BEAST [14], MrBayes [25], RaxML [51], etc. Standard probabilistic fitness models, on the other hand, ignore phylogenetic history and assume that the stationary distribution has been reached,

**Assumption 2.2** (Pure fitness models (FMs)). *Let $\tau_i$ be the distance in time between observed sequence $X_i$ and its parent node. Take $\tau_i \to \infty$ for all $i$, which implies that*

$$X_i \overset{iid}{\sim} \frac{1}{\mathcal{Z}} \exp(\beta f(x)) \text{ for } i \in \{1, 2, \ldots\}. \tag{2}$$

The key implication of this assumption is that density estimation and fitness estimation are linked: the data follows $X_1, \ldots, X_N \sim_{iid} p_0 = p^\infty$, and so if we can estimate $p_0$ we can estimate the fitness. Example models include EVMutation [23], DeepSequence [44], EVE [19], etc. Note although Assumptions 2.1 and 2.2 do not conflict directly, conclusions made based on them conflict in practice: PMs typically infer finite and different lengths for branches (i.e. $\tau_i < \infty$), while FMs typically infer differences in fitness (i.e. $f(x) \neq C$), even when applied to the same dataset.

**1D Example** If Asm. 2.2 does *not* hold, then there is no reason for the distribution of observed sequences $X_1, X_2, \ldots$ to follow $p^\infty$. We illustrate this with an example, the most widely used JFPM that does not use Assumptions 2.1 or 2.2: the Ornstein-Uhlenbeck tree (OUT) model [18, 8]. In this model, $X$ is continuous, i.e. $X \in \mathbb{R}$, and evolves on a quadratic fitness landscape of the form $f(x) \propto (x - \mu)^2 + C$ according to the dynamics $P^\tau(x, x_0) = \text{Normal}\left(x_0 e^{-\frac{1}{2}\tau} + \mu, \sigma^2(1 - e^{-\tau})\right)$. The stationary distribution $p^\infty$ is $\text{Normal}(\mu, \sigma^2)$. One can show (Appx. C.1) that for any $\mathbf{H}$,

**Proposition 2.3** (OUT observations). *The distribution of observed genotypes $X_{1:N}$ is drawn from a multivariate normal distribution with mean $\mu \vec{1}_N$ and covariance $\Sigma$ where*

$$\Sigma_{ij} := \sigma^2 \exp(-\frac{1}{2} t_{ij}(\mathbf{H}))) \text{ for } i, j \in \{1, \ldots, N\}, \tag{3}$$

*and $t_{ij}(\mathbf{H})$ is the total time of the shortest path between leaves $i$ and $j$ along the phylogeny $\mathbf{H}$.*

We drew samples from the OUT with a Kingman coalescent prior on $\mathbf{H}$ ([3] Def. 2.1) and plotted their density (Fig. 1C). Even as $N \to \infty$, the distribution of samples does not follow $p^\infty$. Moreover, rerunning the process with a new sample from the prior yields a very different distribution (Fig. 1D).

# 3 Non-identifiability

In this section we investigate whether, given infinite sequence data, it is possible to infer fitness $f$ without Asm. 2.2, and conversely, whether it is possible to infer phylogeny $\mathbf{H}$ without Asm. 2.1. That is, we are interested in whether fitness and phylogeny are identifiable in general JFPMs. We conclude they are not: given infinite data generated with any $f$ and $\mathbf{H}$, there exists some alternative $\tilde{f}$ and $\tilde{\mathbf{H}}$, where $\tilde{\mathbf{H}}$ satisfies Asm. 2.2, that explains the data equally well.

Naively, this result may be surprising: in FMs, each sequence is drawn independently, i.e. $X_i \perp\!\!\!\perp X_j | \mathbf{H}, f$, while in JFPMs and PMs there is (in general) correlation between sequences, i.e. $X_i \not\perp\!\!\!\perp X_j | \mathbf{H}, f$. One might then hope that examining correlations between sequences would enable us to infer whether Asm. 2.2 holds. However, we can show that these correlations are uninformative due to a symmetry in JFPMs, exchangeability.

**Assumption 3.1** (Exchangeability). *Let $m(X_1, X_2, \ldots)$ denote the marginal probability of an infinite set of sequences $X_1, X_2, \ldots$ integrating over all phylogenies, i.e. $m(X_1, X_2, \ldots) = \int p(X_1, X_2, \ldots | \mathbf{H}) p(\mathbf{H}) d\mathbf{H}$. Then, for any permutation $\pi$ of the integers,*

$$m(X_1, X_2, \ldots) = m(X_{\pi(1)}, X_{\pi(2)}, \ldots). \tag{4}$$

Exchangeability says that if we had observed the sequences in a different order, it would not change their probability. Exchangeability is a ubiquitous assumption in machine learning and statistics models, and its application depends primarily on the information available in a dataset: it is a sensible assumption whenever the ordering of the datapoints provides no useful information. In typical datasets used for fitness estimation, sequences are separated by millions of years of evolution, and are thus all effectively observed at the same time: the present day, $T = 0$. In other words, there is no *a priori* way of ordering the sequences in the dataset, and so we must assume exchangeability. Standard priors on phylogenetic trees, such as the Kingman coalescent, are explicitly constructed to enforce exchangeability [3, 14].

Exchangeability implies that fitness and phylogeny are not identifiable. Even if $X_1, X_2, \ldots$ are generated from a JFPM with a finite branch length phylogeny $\mathbf{H}$, we can describe the same data just as well using an FM model with an infinite branch phylogeny $\tilde{\mathbf{H}}$:

**Theorem 3.2** (Non-identifiability). *Assume $X_1, X_2, \ldots$ satisfy Assumption 3.1. Then with probability 1 there exists some function $\tilde{f}$ such that*

$$X_i \overset{iid}{\sim} p_0(x) = \frac{1}{\tilde{\mathcal{Z}}} \exp(\beta \log \tilde{f}(x)) \ \text{for } i \in \{1, 2, \ldots\}.$$

*Proof.* Applying de Finetti's Theorem ([29], Thm. 11.10), a.s. there exists a random measure $G$ such that for $i \in \{1, 2, \ldots\}$, $X_i \overset{iid}{\sim} G$. Let $p_G(x)$ be the pmf of $G$. (We assume $x$ is a finite discrete sequence; we can also use continuous $x$ assuming the pdf $p_G(x)$ exists.) Set $\tilde{f}(x) = [p_G(x)]^{1/\beta}$. $\square$

This result says that the observed sequences from an exchangeable JFPM, $X_1, X_2, \ldots$, are precisely i.i.d. samples from some $p_0$. Although in the standard tree representation $X_i \not\perp\!\!\!\perp X_j | \mathbf{H}, f$, there must be some alternative description of the same process where $X_i \perp\!\!\!\perp X_j | \tilde{\mathbf{H}}, \tilde{f}$. Fitness and phylogeny are thus non-identifiable: data generated from a JFPM with fitness $f$ and phylogeny $\mathbf{H}$ can be described just as well using $\tilde{f}$ and $\tilde{\mathbf{H}}$, and vice versa. We emphasize that this non-identifiability result is highly general, and does not depend on the specific choice of evolutionary dynamics $P^\tau$, only on the assumption of exchangeability.

The biological intuition behind Thm. 3.2 is that if two sequences are similar to each other and distant from a third, they may be similar either because they are closely related (i.e. the distance $\tau$ to the most recent common ancestor is small) or because they are in a local maximum of the fitness landscape. Without further assumptions, we cannot tell the difference between these two explanations. The machine learning intuition is that evolution, as described by a JFPM, is in effect a Markov chain Monte Carlo process whose stationary distribution gives the fitness. However, the samples we observe may not be fully independent: each pair of samples was initialized from the same point (the most recent common ancestor), and the burn-in since that point may not be sufficiently long. Without independent samples, our estimate of the stationary distribution will be biased.

**Fitness inference as hyperparameter inference** While general, Thm. 3.2 is not constructive, and does not tell us what the distribution $p_0$ actually is, or how exactly it differs from $p^\infty$. Thm. 3.2 also

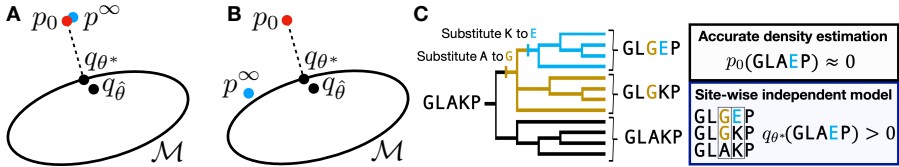

Figure 2: **Alternative explanations for the success of fitness estimation methods.** (A) Setup in which Hypothesis 1 would hold true. (B) Setup in which Hypothesis 2 would hold true. (C) Biological intuition for the benefits of misspecification (Hypothesis 2).

leaves unclear how much we need to know to learn the fitness landscape: could we infer fitness $f$ if we knew the parametric form of $p^\infty$, i.e. if we had some model $\mathcal{M}$ and knew that $p^\infty \in \mathcal{M}$? What if we also knew the underlying phylogeny $\mathbf{H}$? In the long branch limit (Asm. 2.2), fitness is identifiable if $\mathbf{H}$ is known; if $\mathcal{M}$ is also known, learning fitness is a matter of inferring model parameters. In the limit where all the branch lengths in the phylogeny are zero, the distribution of observations from a JFPM reduces to $X_1 \sim p_\infty$ and $X_1 = X_2 = X_3 = \ldots$. Here fitness is non-identifiable even if $\mathbf{H}$ and $\mathcal{M}$ are known; learning fitness is a matter of learning from a single sample. In the realistic intermediate branch length case, if $\mathbf{H}$ and $\mathcal{M}$ are known, we will show that learning fitness is essentially a matter of *hyperparameter* rather than *parameter* inference.

We demonstrate this last claim in the context of the OUT example, by approximating the OUT model as a Gaussian process latent variable model (GPLVM). We find that fitness only appears as a hyperparameter of the derived GP, not as a parameter. The GPLVM has latent variables $Z_1, Z_2, \ldots$ that lie on the hyperbolic plane $\mathbb{H}$, and uses the Gaussian process kernel $k(\cdot, \cdot) = \exp(-d(\cdot, \cdot))$, where $d(\cdot, \cdot)$ is a distance metric over $\mathbb{H}$. Let $\mathcal{W}_1(\cdot, \cdot)$ be the Wasserstein metric for distributions over infinite matrices, i.e. over $\mathbb{R}^{\infty \times \infty}$, using the sup norm on matrices.

**Theorem 3.3** (GPLVM approximation of OUT). *Assume a prior over phylogenies $\mathbf{H}$ that is exchangeable in its leaves and where the minimum time between any pair of nodes is greater than $\eta > 0$ with probability 1. Define the leaf distance matrix $\nu_{ij} = \log(\frac{1}{2} t_{ij}(\mathbf{H}))$. For any $\epsilon > 0$, there exists a.s. a GPLVM of the form,*

$$G \sim \mathcal{G}, \qquad s \sim \text{GaussianProcess}(\mu, \sigma^2 k(\cdot, \cdot)),$$
$$Z_i \overset{iid}{\sim} G \text{ for } i \in \{1, 2, \ldots\}, \tag{5}$$
$$X_i = s(Z_i),$$

*where $G$ is a random measure over $\mathbb{H}$, such that $\mathcal{W}_1(p(\nu), p(\tilde{\nu})) < \epsilon$, where $\tilde{\nu}_{ij} = \log(d(Z_i, Z_j))$.*

*If $\mathcal{W}_1(p(\nu), p(\tilde{\nu})) = 0$, the OUT and GPLVM produce identical distributions over $X_1, X_2, \ldots$ a.e..*

The proof is in Appx. C.2, and uses the embedding of Sarkar [48]. This result says that, by embedding phylogenies $\mathbf{H}$ in a metric space, we can approximate an OUT arbitrarily well with a GPLVM; as the Wasserstein bound gets smaller, the distribution of covariance matrices of the two models get closer. In the GPLVM, the observations are conditionally independent, $X_i \perp\!\!\!\perp X_j | s, G$, in line with Thm. 3.2. The phylogeny $\mathbf{H}$ enters the GPLVM only through the latent space embedding $Z_1, Z_2, \ldots$. Learning phylogeny, given the fitness landscape, is thus essentially a matter of inferring latent variables [44, 13]. The fitness landscape enters the GPLVM only through the prior on the Gaussian process (i.e. through $\mu$ and $\sigma$). Inferring fitness given phylogeny is thus essentially a matter of inferring hyperparameters. This is both good and bad news for fitness inference. On the one hand, hyperparameters are often learned in practice, and doing so can yield substantially better predictions, so we should be able learn something about $\mu$ and $\sigma$ given data ([58], Chap. 5). On the other hand, hyperparameters are in general (though not always) non-identifiable, and therefore so is fitness [34]. Ho and Ané [22] describe non-identifiability conditions for the OUT in particular. We conclude that even when $\mathbf{H}$ and $\mathcal{M}$ are known, fitness inference in JFPMs is fundamentally challenging.

## 4 Benefits of misspecification

We have demonstrated that a plausible biological mechanism – namely, phylogenetic effects – can produce a data distribution $p_0$ that does not reflect fitness, and can make exact inference of fitness impossible even given infinite data. Nonetheless, the practical success of fitness estimation methods suggest it is possible to at least approximate fitness landscapes from observational sequence data.

Recall that existing methods proceed by fitting a probabilistic model $q_\theta \in \mathcal{M} = \{q_\theta : \theta \in \Theta\}$ to data $X_{1:N}$, typically via maximum likelihood estimation or approximate Bayesian inference, and then using the predicted log density $\log q_{\hat{\theta}}(x)$ as an estimate of the fitness of a sequence $x$. Why is this approach empirically successful? In this section we consider two hypotheses, either of which may hold true in theory. In Secs. 6–7 we develop and apply tests to evaluate them on real data.

Note that our results in this and the following sections are independent of the specific evolutionary mechanisms that generate a mismatch between $p_0$ and $p^\infty$, i.e. they are not specific to phylogenetic effects or JFPMs, nor do they even depend on the existence of a stable reproductive fitness function $f$ over evolutionary time. We can, in fact, redefine $p^\infty$ to be an arbitrary "target distribution", with $\log p^\infty$ proportional to any chosen measure of molecular fitness or function (such as enzyme activity, fluorescence, etc.). For the sake of concrete illustration, however, we will continue to focus on JFPMs as our primary example of why the data distribution, $p_0$, may not equal the target distribution we want to estimate, $p^\infty$.

We consider two hypotheses for the empirical success of existing fitness estimation methods.

**Hypothesis #1** (informal). *Fitness estimation methods succeed by finding $q_{\hat{\theta}} \approx p_0$, since for all practical purposes on real data, $p_0 = p^\infty$.*

This hypothesis would make sense, in JFPMs, if Asm. 2.2 held, i.e. branch lengths were long enough in real datasets for $P^{\tau_i}(x, x_0)$ to be close to its stationary distribution. Under this hypothesis, better density estimators have been, and will continue to be, better fitness estimators. We should focus on developing models $\mathcal{M}$ that are well-specified with respect to the data, i.e. $p_0 \in \mathcal{M}$ (Fig. 2A).

**Hypothesis #2** (informal). *Fitness estimation methods succeed by using models $\mathcal{M}$ that are misspecified with respect to $p_0$, i.e. $p_0 \notin \mathcal{M}$. The inferred model $q_{\hat{\theta}}$ is then closer to $p^\infty$ than $p_0$ is.*

To show this second hypothesis is plausible, we prove that it is guaranteed to hold under general conditions. We study the projection of $p_0$ onto $\mathcal{M}$ via the Kullback-Leibler (KL) divergence, $q_{\theta^*} = \operatorname{argmin}_{q_\theta \in \mathcal{M}} \operatorname{KL}(p_0 \| q_\theta)$. The KL projection is relevant because maximum likelihood estimation minimizes the approximate KL divergence between the data and the model, and the posterior in Bayesian inference asymptotically concentrates around the maximum likelihood estimator [38]. We thus expect the fit model $q_{\hat{\theta}}$ to be close to $q_{\theta^*}$, and get closer with $N$. Assume that $\mathcal{M}$ is "log-convex", meaning that for any $\theta, \theta' \in \Theta$ and $0 < r < 1$, there exists some $\theta''$ such that $q_{\theta''}(x) = q_\theta(x)^r q_{\theta'}(x)^{1-r} / \sum_x q_\theta(x)^r q_{\theta'}(x)^{1-r}$; examples of log-convex models include the Potts model, as well as all other exponential family models. Let $\operatorname{TV}(p \| q)$ be the total variation distance between $p$ and $q$, and let $\|g\|_\infty = \sup_x |g(x)|$ be the uniform (sup) norm of $g$.

**Theorem 4.1** (Benefits of misspecification). *Assume that the model $\mathcal{M}$ is log-convex and that $q_{\theta^*}$ exists and is unique. If the model is "less misspecified" with respect to the stationary distribution $p^\infty$ than with respect to the data distribution $p_0$, in the sense that*

$$\min_{q_\theta \in \mathcal{M}} \| \log q_\theta - \log p^\infty \|_\infty < \operatorname{TV}(q_{\theta^*} \| p_0), \tag{6}$$

*then,*

$$\operatorname{KL}(q_{\theta^*} \| p^\infty) < \operatorname{KL}(p_0 \| p^\infty). \tag{7}$$

*However, if the model is well-specified with respect to the data distribution, i.e. $p_0 \in \mathcal{M}$, we have,*

$$\operatorname{KL}(q_{\theta^*} \| p^\infty) = \operatorname{KL}(p_0 \| p^\infty). \tag{8}$$

Sec. C.3 contains the proof, and explains how Thm. 4.1 can be extended with more general conditions (we also emphasize again that the proof does not make any assumption that $p_0$ and $p^\infty$ follow a JFPM). Thm. 4.1 says that if the model is less misspecified with respect to the target distribution than with respect to the data distribution, then projecting the data distribution onto the model will yield a distribution $q_{\theta^*}$ closer to the target distribution. In other words, the best models for fitness estimation are those at a "sweet spot" of complexity, flexible enough to capture $p^\infty$ but not so flexible as to capture $p_0$.

To understand the biological intuition behind this result, consider a situation where two neutral mutations with no effect on fitness occur successively at different sites (Fig. 2C). Due to phylogenetic correlation, there is no observed sequence $x^*$ in which the second mutation is present but not the first, so an accurate density estimator will find $p_0(x^*) \approx 0$. However, if we can guess correctly that the fitness landscape is independent across sites, then fitting a site-wise independent model $\mathcal{M}$ will imply the mutation is allowed, $q_{\theta^*}(x^*) > 0$, correctly inferring $p^\infty(x^*) > 0$.

Under Hypothesis 2, progress in the field of fitness estimation has *not* come from building better density estimators (Hypothesis 1), but rather from an iterative process of (1) hypothesizing, based partly on biophysical knowledge, models that are approximately well-specified with respect to $p^\infty$ but poorly specified with respect to the data distribution $p_0$, and then (2) comparing their density estimates against experimental fitness measurements. We will show that on real data, Hypothesis 1 can often be rejected in favor of Hypothesis 2.

## 5   Related Work

Efforts to account for the effects of phylogeny in fitness estimation have a long history [32]. Practical generative sequence models that explicitly account for both epistatic fitness landscapes and phylogeny have long been sought, but stymied primarily by computational challenges [28, 46]. In their place, a variety of non-generative (and often heuristic) methods for correcting for phylogeny have been proposed, including data reweighting schemes [35, 46], data segmentation schemes [10], post-inference parameter adjustments [16], covariance matrix denoising methods [42], simulation based statistical testing [45], and more. In this article, we show that deconvolving fitness and phylogeny is not just computationally hard, but also in general statistically impossible: fitness and phylogeny are non-identifiable. We further show that use of a misspecified parametric model can on its own (without further corrections) partially adjust for phylogenetic effects.

Our results also intersect with the literature on robust statistics: we can think of the observed data distribution $p_0$ as a "distorted" version of the true distribution of interest $p^\infty$. However, in typical robust inference frameworks (e.g. Huber's epsilon contamination model), the observed distribution differs from the true distribution by the addition of outliers [24, 52]. Our setup is, in some sense, the opposite: inliers are deleted, as phylogenetic correlations can result in an effective support of $p_0$ that is *smaller* than that of $p^\infty$ (Fig. 1CD).

## 6   Diagnostic Method

In this section, we develop diagnostic methods to discriminate between Hypothesis 1 and Hypothesis 2 (Sec. 4) based on observational sequence data and experimental fitness measurements, and validate these diagnostics in simulation. Recall that under Hypothesis 2, the estimate $q_{\hat\theta}$ from a parametric fitness model is a better estimate of fitness than the true data density $p_0$, while under Hypothesis 1, $p_0$ is better. Discriminating these two hypotheses on real data is nontrivial because we do not have access to $p_0$. Ideally, then, a diagnostic test would evaluate the probability that the true density $p_0$ outperforms $q_{\hat\theta}$ at predicting fitness, taking into account uncertainty in what $p_0$ could actually be, given the data. To accomplish this, we compute a posterior over $p_0$ using a Bayesian nonparametric sequence model. In particular, we apply the Bayesian embedded autoregressive (BEAR) model, which can be scaled to terabytes of data and satisfies posterior consistency ([2], Thm. 35):

**Theorem 6.1** (Summary of BEAR posterior consistency). *Assume $p_0$ is subexponential, i.e. for some $t > 0$, $\mathbb{E}_{X \sim p_0}[\exp(t|X|)] < \infty$, where $|X|$ is the length of sequence $X$. Assume the conditions on the prior detailed in Amin, Weinstein and Marks [2]. If $X_1, X_2, \ldots \sim p_0$ i.i.d., then for $M > 0$ sufficiently large and $\epsilon \in (0, 1/2)$ sufficiently small,*

$$\Pi_{\mathrm{BEAR}}(B(p_0, MN^{-\epsilon})|X_{1:N}) \xrightarrow{N \to \infty} 1$$

*in probability, where $B(p, r)$ is a Hellinger ball of radius $r$ centered at $p$, and $\Pi_{\mathrm{BEAR}}(\cdot|X_{1:N})$ is the BEAR posterior.*

Crucially, this result implies that the BEAR posterior will converge to effectively any value of $p_0$, no matter what $p_0$ is (unlike a parametric model's posterior). Moreover, BEAR quantifies uncertainty in its estimates, giving the range of possible values of $p_0$ that are consistent with the evidence.

We construct our diagnostic test by comparing the fitness estimation performance of $q_{\hat\theta}$ to the range of possible performances of $p_0$ estimated by BEAR. Let $\mathcal{S}_f(p)$ be a scalar score evaluating how accurately a density $p$ predicts fitness $f$. In practice, $\mathcal{S}_f$ will be based on experimental and clinical measurements of quantities directly related to fitness.

**Diagnostic test** (Test Hypothesis 1 vs. Hypothesis 2.)   *Hypothesis 1 $\mathcal{H}_1 : \mathcal{S}_f(q_{\hat\theta}) < \mathcal{S}_f(p_0)$. Hypothesis 2 $\mathcal{H}_2 : \mathcal{S}_f(q_{\hat\theta}) > \mathcal{S}_f(p_0)$. Accept Hypothesis 2 at significance level $\alpha > 0$ if*

$$\Pi_{\mathrm{BEAR}}(\mathcal{S}_f(q_{\hat\theta}) > \mathcal{S}_f(p)|X_{1:N}) > 1 - \alpha. \tag{9}$$

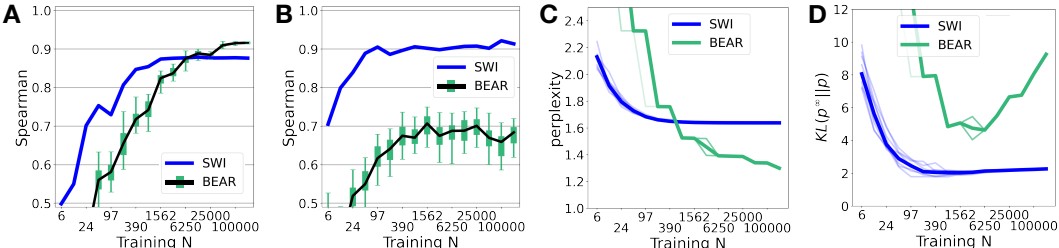

Figure 3: **BEAR diagnostic applied to simulated data.** (A) Scenario 1. Spearman correlation between the maximum likelihood SWI model and the true fitness $\mathcal{S}_f(q_{\hat{\theta}})$, compared to the BEAR posterior distribution over $\mathcal{S}_f(p)$. Quantiles and 95% credible interval shown with green box and whisker. Points above (below) the whiskers correspond to SWI models that significantly outperform (underperform) the true data distribution. (B) Same as A, for Scenario 2. (C) Perplexity on heldout data of the BEAR and the SWI models in Scenario 2. Thick line corresponds to the average over 10 individual simulations (thin lines). (D) Same as C, comparing the KL divergence to $p^{\infty}$.

*Accept Hypothesis 1 at significance level $\alpha$ if*

$$\Pi_{\text{BEAR}}(\mathcal{S}_f(q_{\hat{\theta}}) < \mathcal{S}_f(p)|X_{1:N}) > 1 - \alpha. \tag{10}$$

So long as $\mathcal{S}_f(p)$ is a well-behaved function of $p$ (in particular, so long as $\mathcal{S}_f$ is continuous in a neighborhood of $p_0$ with respect to the topology of convergence in total variation), Thm. 6.1 implies that this diagnostic test will be asymptotically consistent, in the sense that it converges to the correct hypothesis in probability.

**Simulations** We next evaluate the performance of our diagnostic test on simulated data. We considered two scenarios, the first in which Hypothesis 1 holds, and the second in which Hypothesis 2 holds. In both, we let $\mathcal{M}$ be a site-wise independent (SWI) model, in which each position of the sequence is drawn independently, i.e. $X_l \sim \text{Categorical}(v_l)$ for $l \in \{1, \ldots, |X|\}$. The parameter $v_l$ is in the simplex $\Delta_B$, where $B + 1$ is the alphabet size. (Further details in Appx. D.) In Scenario 1, the true data are generated according to a Potts model and $p_0 = p^{\infty}$. In this scenario, the SWI model is misspecified, and misspecification is *bad*: using a more flexible model will produce an asymptotically more accurate estimate of $p^{\infty}$. We find that our diagnostic test asymptotically correctly accepts Hypothesis 1, in line with Thm. 6.1 (Figs. 3A and 7A). In Scenario 2, the true data are generated according to a JFPM with finite branch lengths, and $p^{\infty} \in \mathcal{M}$ while $p_0 \notin \mathcal{M}$. The mutational dynamics $P^{\tau}$ follow the Sella and Hirsh [49] process. The phylogeny $\mathcal{H}$ is drawn from a Kingman coalescent. In this scenario, the SWI model is again misspecified, but misspecification is *good*: while the nonparametric BEAR model can achieve better density estimates than the SWI model (Fig. 3C), the SWI model outperforms BEAR at fitness estimation (Figs. 3D and 8). We find that our diagnostic test correctly accepts Hypothesis 2 (Figs. 3B and 7B).

A possible point of concern is that the test is poorly calibrated from a frequentist perspective, and in the low $N$ regime accepts Hypothesis 2 in Scenario 1 more than $100\alpha\%$ of the time when the data is resampled from $p_0$ (Fig. 9A). This behavior is common in nonparametric Bayesian tests, and not necessarily a problem: the test is still valid from a purely Bayesian perspective. Nevertheless, on real data we will check that we are close to the large $N$ regime by (1) checking that the BEAR posterior predictive is at least as close to $p_0$ as $q_{\hat{\theta}}$ is (as measured by perplexity on held out data; Figs. 3C and 9B) and (2) examining the plot of the BEAR posterior over $\mathcal{S}_f(p)$ as a function of $N$ (as in Fig. 3AB), to check that it has converged.

## 7 Empirical Results

We now evaluate whether existing fitness estimation methods outperform the true data density $p_0$, i.e. whether we can reject Hypothesis 1 in favor of Hypothesis 2 on real data.

**Tasks** We consider two key prediction tasks where fitness models are applied in practice. The first task is to predict whether variants of a protein are functional, according to an experimental assay of protein function; the metric $\mathcal{S}_f(\cdot)$ is the Spearman correlation between $p(x)$ and the assay result [23]. There are typically ~1000s of measurements per assay. The second task is to predict whether a variant of a protein observed in humans causes disease, according to clinical annotations; the metric $\mathcal{S}_f(\cdot)$ is the area under the ROC curve when $p(x)$ is used to predict whether or not a variant is

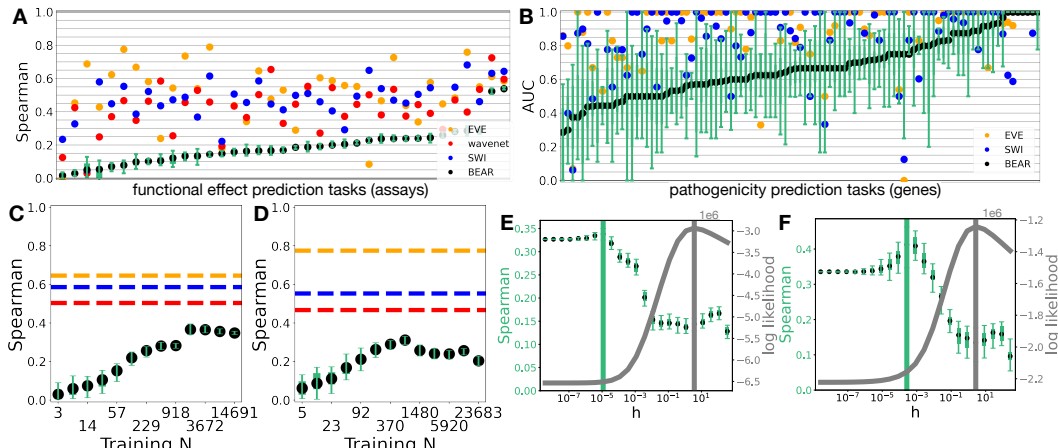

Figure 4: **Fitness estimation models systematically outperform the data distribution.** (A) Results for the first prediction task, predicting functional measurements in experimental assays. Quantiles and 95% credible interval of the BEAR posterior are shown with the green box and whisker plot. Points above (below) the whiskers correspond to fitness estimation models that significantly outperform (underperform) the true data distribution. (B) Results for the second prediction task, predicting variant pathogenicity in human genes. (C) Convergence of the BEAR posterior with datapoints $N$, for an example assay ($\beta$-lactamase). (D) Same as C, for another example assay (TIM barrel). (E) BEAR posterior Spearman (black and green) versus BEAR log likelihood (gray), interpolating between parametric and nonparametric regimes (low and high $h$), for an example assay (another $\beta$-lactamase assay). Peak Spearman indicated with vertical green line, peak log likelihood with gray. (F) Same as E, for another example assay (GAL4 DNA-binding domain).

pathogenic [19]. There are typically only a handful of labels for each gene. For the first task, we considered 37 different assays across 32 different protein families, and for the second task, 97 genes across 87 protein families; for each protein family, we assembled datasets of evolutionarily related sequences, following previous work. Note that across the 37 assays and 97 genes, the data used for $\mathcal{S}_f$ comes from different experiments and different clinical evidence, often collected by different laboratories or doctors. Thus, our overall conclusions should be robust to the choice of $\mathcal{S}_f$.

**Models** We considered three existing fitness estimation models: a site-wise independent model (SWI), a Bayesian variational autoencoder (EVE [19], which is similar to DeepSequence [44]), and a deep autoregressive model (Wavenet) [50]. Note that SWI and EVE, unlike Wavenet, require aligned sequences as training data. Details in Appx. E.

**Results** Applied to the first prediction task, our diagnostic test accepts Hypothesis 2 at significance level $\alpha = 0.025$ in 35/37 assays (95%) for SWI, 35/37 assays (95%) for EVE, and 36/37 assays (97%) for Wavenet (Fig. 4A). Applied to the second prediction task, our diagnostic test accepts Hypothesis 2 at significance level $\alpha = 0.025$ in 31/97 genes (32%) for SWI and 46/97 genes (47%) for EVE (Fig. 4B). Thus, fitness estimation models are capable of outperforming the true data distribution $p_0$. We found evidence for Hypothesis 1 in only a handful of examples: on the first task, Hypothesis 1 was accepted at significance level $\alpha = 0.025$ in 0/37 assays for SWI, 1/37 assays (3%) for EVE, and 0/37 assays for Wavenet, while on the second task, Hypothesis 1 was accepted for 5/97 genes (5%) for SWI and 4/97 genes (4%) for EVE. We confirmed that the diagnostic test was in the large $N$ regime: BEAR outperformed Wavenet at density estimation, providing better predictive performance on 27/37 assays (73%) and similar performance on the remaining 10 assays (Fig. 10). (Note that we cannot do this comparison for SWI or EVE since they are alignment-based [57].) Example plots of the BEAR posterior's convergence with $N$ on the first prediction task showed convergence to values of $\mathcal{S}_f$ well below that for parametric fitness estimation models (Figs. 4C and 11-12). Overall, we conclude that there is strong evidence that existing fitness estimation methods reliably outperform the true data distribution $p_0$ across a range of datasets and tasks.

To study the tradeoffs between density estimation and fitness estimation in more depth, we smoothly and nonparametrically relaxed a parametric autoregressive (AR) model (Appx. E.4). We embedded the AR model (a convolutional neural network) into a BEAR model, and fit the BEAR model with empirical Bayes. We found evidence that the AR model was misspecified on every dataset, following the methodology of Amin, Weinstein and Marks [2]: the optimal $h$ selected by empirical Bayes

was on the order of $1 - 10$ in each dataset. Now, in the limit as the hyperparameter $h \to 0$, the BEAR model collapses to its embedded AR model; so by scanning $h$ from low to high values we can interpolate between the parametric and nonparametric regime. We find a smooth tradeoff between $\mathcal{S}_f(p)$ and the likelihood of the data under the BEAR model, with higher $h$ corresponding to better density estimation but worse fitness estimation (Fig. 4EF and 13). This relationship held across many datasets: the diagnostic test, evaluated against the AR model (the $h \to 0$ limit), accepts Hypothesis 2 in 28/37 assays (76%), but Hypothesis 1 in only 6/37 (16%) (Fig. 14). These results confirm that making a model well-specified (relaxing from a parametric to a nonparametric model) can bring improved density estimation at the cost of worse fitness estimation.

## 8   Discussion

In this article, we have argued that better density estimation does not necessarily lead to better fitness estimation. Further, we estimate with high probability that existing fitness estimation methods systematically outperform the true training data density. Although existing methods rely on flexible, high-parameter deep neural network models, they can nonetheless be misspecified; but this misspecification acts as a blessing, rather than a curse for fitness estimation. Successful models are at a sweet spot of complexity, flexible enough to capture the target fitness distribution well but not so flexible as to match the data distribution itself.

We have focused on state-of-the-art fitness estimation methods which are trained on data from individual protein families [44, 50, 19]. Recently, large-scale generative sequence models ("protein language models") trained on more diverse datasets (containing proteins from many different families) have show fitness estimation performance comparable to, and in some settings surpassing, single family models [36, 40, 41]. Although applying our diagnostic test to these datasets requires further work, there is no reason to expect the same limitations of density estimation do not hold for such models. Indeed, following a preprint of this paper, Nijkamp et al. [40] presented evidence of the benefits of misspecification in a protein language model: past a certain number of parameters, density estimation improved while fitness estimation deteriorated. See Appx. G for further discussion.

One future direction is to further explore models $\mathcal{M}$ that are *less* flexible than existing models and *worse* at density estimation, since they can increase the gap between $\mathrm{KL}(q_{\theta*}\|p^\infty)$ and $\mathrm{KL}(p_0\|p^\infty)$ (Thm. 4.1). There may also be opportunity to improve model geometry: while exponential family models are guaranteed to be log-convex (and thus can satisfy Thm. 4.1), we have no such guarantee for variational autoencoders or other neural network methods. Meanwhile, uncertainty quantification is crucial for applications such as those in clinical genetics, but challenging in misspecified models [53, 39, 26]. Alternatively, it might be useful to abandon the strategy of using misspecified models for fitness estimation altogether, and instead construct JFPM models where the fitness landscape is represented explicitly as a latent variable. Recent progress on amortized variational inference for phylogenetic models is promising for building flexible and scalable JFPMs [55]. However, handling non-identifiability is challenging, and may require new assumptions and/or new methods of sensitivity analysis to infer the full set of fitness landscapes consistent with the data [9].

Although this article has focused on technological applications of fitness models in solving prediction problems, fitness models also have implications for our fundamental understanding of evolution. Pure phylogeny models and pure fitness models present very different pictures of the past history of life: in PMs, similarities and differences among genetic sequences are determined primarily by history and ancestry (Asm. 2.1), while in FMs they are primarily determined by functional constraints (Asm. 2.2). PMs and FMs also present very different implications for the future of life: in PMs, the diversity of sequences seen in nature will likely expand dramatically going forward, while in FMs, the landscape of functional sequences has already been well-explored. Our results emphasize that where and to what extent each model offers an accurate picture of reality remains an open question.

### Acknowledgements

We thank members of the Marks lab, Pascal Notin, Ali Madani, John Ingraham and the anonymous reviewers for insights and suggestions. E.N.W.'s work was supported by the Fannie and John Hertz Foundation. D.S.M. is supported by the Chan Zuckerberg Initiative.

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
