# OpenReview forum: "Non-identifiability and the Blessings of Misspecification in Models of Molecular Fitness"
_NeurIPS.cc/2022/Conference — NeurIPS 2022 Accept_

### Official Review · Reviewer_THKS · 2022-07-11

**Rating:** 7
**Confidence:** 4
**Soundness:** 3 good
**Presentation:** 4 excellent
**Contribution:** 2 fair

**Summary:**

The paper asks two important questions in estimating molecular fitness using generative models for sequences: 1) Is fitness fully identifiable from the observation of sequence data alone? and 2) Is developing generative models that best emulate the data distribution necessary or sufficient to estimate fitness? Under certain assumptions and statistical model for sequences and fitness (which is later validated on real-data), the paper demonstrate that the answer to both questions (perhaps surprisingly) is no, i.e., 1) Fitness is not fully identifiable even from infinitely many number of observed sequences and 2) Generative models are successful in predicting fitness not necessarily because they fit the original data distribution the best, rather because they are trained over a function class (e.g., Potts models or Neural networks) that better fit to another so-called stationary distribution of sequence which originates from phylogenetic models and this captures fitness in the long run.

**Questions:**

How large should the sequence dataset be for fitness estimation? The paper shows that better estimates of p0 using infinitely many sequence data is not the best strategy to model fitness since we like to estimate p\infinity instead. Obviously a very small amount of sequence data is also not useful for fitness estimation. This seems to suggest there is a sweet spot in terms of the amount of sequence data useful for fitness estimation after which there will get into a diminishing return regime. Is there any way to find what the sweet spot is perhaps using the analysis done in Fig 2? This will help since we will know how much sequence data is operationally enough for practical purposes and also give insights on how big and complex a model is sufficient for generative modeling towards the goal of fitness estimation.

Assumptions.  The analysis of identifiability of fitness naturally relies on some model for sequence and fitness function. Here, an Ornstein-Uhlenbeck tree model is employed in JFPM which evolves over a quadratic fitness landscape and a Kingman coalescent prior over the tree. While this particular choice completely shows an example of a case where fitness is not identifiable from the sequence data alone, is it fair to draw a universal conclusion for the rest of the models not considered here or fitness functions for which we don't have a good model for? Similarly another assumption seems to be the existence of a stationary p\infinity distribution that in the long run converges to an energy landscape corresponding to fitness. Is this always the case for all molecules and fitness functions? Can we imagine molecules with multiple fitness measures, some of which we are interested to estimate however are more neutral to the selection process and some others that we are not interested to measure however are more influential in selection? E.g., fluorescence (as objective fitness) versus thermostability (driving selection) of a protein.

Family of misspecified models for M. This is suggested in the future work but really seems to strengthen the exposition of the paper to give some demonstrations of potential models that are worse at density estimation and are better at fitness estimation. While the hypothesis testing procedure is a valid way to convey the message of the paper on misspecification, developing such M models would be the ultimate constructive way to improve the state of the art on fitness estimation while justifying the misspecification claim.

Transformers. Since different variants of transformers and other large-scale language models are now at the core of generative modeling of sequences it would be beneficial to bring a summary of the discussions from Appendix G to the paper. How much do we expect the two messages of the paper hold for protein language models?

**Limitations:**

Discussed before.

**Strengths And Weaknesses:**

Overall the questions posed by the authors are interesting and important and have clear messages that help the community of molecular inference. The approach to address the questions are clearly described, statistically sound, and opens new research questions. For these reasons I think the paper is above the acceptance line. Perhaps the key comment is that the analysis largely relies on certain assumptions on sequences and fitness which might restrict the scope of the paper. Furthermore, the final message of the paper is more of a statistical property of the biological sequence data and fitness; the paper can be strengthened by some constructive and operational algorithms inspired by the key message of the paper to help better model fitness. I will expand on these comments here.

---

> ### Author Response · Authors · 2022-08-01
> **Response to Reviewer THKS Part 1**
>
> We thank the reviewer for their thoughtful response.
>
> “How large should the sequence dataset be for fitness estimation? The paper shows that better estimates of p0 using infinitely many sequence data is not the best strategy to model fitness since we like to estimate p\infinity instead. Obviously a very small amount of sequence data is also not useful for fitness estimation. This seems to suggest there is a sweet spot in terms of the amount of sequence data useful for fitness estimation after which there will get into a diminishing return regime. Is there any way to find what the sweet spot is perhaps using the analysis done in Fig 2? This will help since we will know how much sequence data is operationally enough for practical purposes and also give insights on how big and complex a model is sufficient for generative modeling towards the goal of fitness estimation.”
>
> To address the question of how large a sequence dataset should be used, it is useful to distinguish between the case where the model is misspecified and where the model is well specified. In the case where the model is misspecified, we expect it to converge to $q_{\theta^*}$ as the amount of data increases to infinity. Under our hypothesis 2, $q_{\theta^*}$ is a reasonable (though not perfect) fitness estimator, and so increasing the amount of data will in general help performance (at least up to a point). Nonparametric models are expected to converge to $p_0$ in the large data limit. However, on finite data, the model's prior (or any other form of regularization) can have important effects. If the prior is close to  $p^\infty$, it may be the case that (as the reviewer proposes) there is a sweet spot for the amount of data to train on for fitness estimation. The sweet spot corresponds to a compromise between the information provided by the prior and the information provided by the data. Such a sweet spot can be seen in Figure 4CD, which shows the fitness estimation performance of a nonparametric model (BEAR) as a function of data. The position of the sweet spot (the best amount of data) will in general depend on the exact dataset, model and prior/regularization. Since our analysis concludes that Hypothesis 2 holds and misspecified models can be better fitness estimators then well specified/nonparametric models, we recommend using datasets that are as large as possible. As can be seen from Figure 4CD, even if we somehow could choose perfectly the sweet spot for a regularized nonparametric model, it is unlikely to outperform a misspecified model fit to a large dataset.
>
>
> "Assumptions. The analysis of identifiability of fitness naturally relies on some model for sequence and fitness function. Here, an Ornstein-Uhlenbeck tree model is employed in JFPM which evolves over a quadratic fitness landscape and a Kingman coalescent prior over the tree. While this particular choice completely shows an example of a case where fitness is not identifiable from the sequence data alone, is it fair to draw a universal conclusion for the rest of the models not considered here or fitness functions for which we don't have a good model for? ”
>
> Our proof that fitness and phylogeny are non-identifiable (Theorem 3.2) rests on extremely general assumptions (Assumption 3.1) and does not depend on the specific choice of model or fitness function. Our results on the Ornstein-Uhlenbeck model (Theorem 3.3) serve to illustrate our abstract results with a concrete and informative example. Indeed, since fitness and phylogeny are non-identifiable even in the extremely simple case of an Ornstein-Uhlenbeck model with Kingman coalescent prior, this gives us confidence that there is no obvious, plausible assumption that could be added to Assumption 3.1 that would result in identifiability. In other words, Theorem 3.3 shows that our general non-identifiability result (Theorem 3.2) is robust to strong simplifying assumptions.

---

> ### Author Response · Authors · 2022-08-01
> **Response to Reviewer THKS Part 2**
>
> "Similarly another assumption seems to be the existence of a stationary p\infinity distribution that in the long run converges to an energy landscape corresponding to fitness. Is this always the case for all molecules and fitness functions? Can we imagine molecules with multiple fitness measures, some of which we are interested to estimate however are more neutral to the selection process and some others that we are not interested to measure however are more influential in selection? E.g., fluorescence (as objective fitness) versus thermostability (driving selection) of a protein."
>
> Fitness estimation from evolutionary sequence data is only expected to work if the phenotype of interest is under selection. Otherwise, there is no reason for the data distribution (by itself) to contain any information about the phenotype. However, fitness estimation using misspecified generative models does not depend on the existence of a well-defined stationary distribution for evolution. Rather, it depends only on the assumptions that (a) there exists some distribution function $p^\infty(x)$ we can define that is correlated with the results of the assay we are interested in, (b) that the data we have comes from a distribution $p_0$ that is informative of $p^\infty$ but does not match $p^\infty$ exactly, and (c) that the difference between $p_0$ and $p^\infty$ is similar across different assays, such that we can choose a single model class M that works well for many or all of the different assays. Note, in particular, that our proof of the blessings of misspecification (Theorem 4.1) does not depend on any evolutionary assumptions about where $p^\infty$ comes from.
> The theoretical analysis of fitness and phylogeny in Sections 2-3 is meant to provide a plausible biophysical mechanism explaining why the observed data distribution might not match the distribution of interest, and why it might be fundamentally hard to estimate the distribution of interest. If selective pressures fluctuate over time, or if selective pressures are not on the trait being measured in the assay but rather on a related trait, this can also lead to a situation where the data distribution does not reflect the distribution of interest exactly. In this case, Theorem 4.1 can still apply, and fitting misspecified generative models may still provide useful predictions of experimental results.
>
>
> "Family of misspecified models for M. This is suggested in the future work but really seems to strengthen the exposition of the paper to give some demonstrations of potential models that are worse at density estimation and are better at fitness estimation. While the hypothesis testing procedure is a valid way to convey the message of the paper on misspecification, developing such M models would be the ultimate constructive way to improve the state of the art on fitness estimation while justifying the misspecification claim.”
>
> In Figures 4 and 10, we show that the Wavenet model is worse at density estimation but better at fitness estimation, in comparison to a nonparametric density estimator (BEAR). The same holds for the autoregressive neural network embedded in BEAR, as can be seen from Figure 13. Although it is an MSA-based model and so we cannot evaluate its density estimation performance directly, the sitewise independent model (SWI) has a very small number of parameters and does strikingly better at fitness estimation than a nonparametric model (BEAR).
>
>
> "Transformers. Since different variants of transformers and other large-scale language models are now at the core of generative modeling of sequences it would be beneficial to bring a summary of the discussions from Appendix G to the paper. How much do we expect the two messages of the paper hold for protein language models?”
>
> Theoretically, there is no reason to expect our results to change when applied to models trained on large datasets. Empirically, Nijkamp E et al, 2022 recently presented evidence that the blessings of misspecification which we describe indeed hold for large-scale transformers trained on datasets consisting of sequences from across evolution. They showed in particular that increasing model size past 1 billion parameters led to better density estimation but worse fitness estimation. We will add a summary of Appendix G and the results of Nijkamp et al. to the paper.
> Citation:
> ProGen2: Exploring the Boundaries of Protein Language Models; Nijkamp, Erik and Ruffolo, Jeffrey and Weinstein, Eli N. and Naik, Nikhil and Madani, Ali; 2022; arXiv

---

> > ### Comment · Reviewer_THKS · 2022-08-07
> > **Thanks**
> >
> > Thanks for the response. I don't have further questions and willing to increase my score to 7 based on the feedback and the planned edits to the manuscript.

---

### Official Review · Reviewer_wgyk · 2022-07-11

**Rating:** 7
**Confidence:** 4
**Soundness:** 4 excellent
**Presentation:** 4 excellent
**Contribution:** 3 good

**Summary:**

The authors illuminate an issue with generative models fit to protein sequence data. Often, the likelihoods derived from these models are interpreted as a proxy for evolutionary fitness and applied on tasks like mutation pathogenicity prediction. However, theses models are mid-specified when they don’t explicitly consider the evolutionary process. The authors produce mathematical explanations of this mid-specification and explore various hypotheses with simulated and real data. The argument is clear and compelling. I appreciate the authors highlighting paths forward, for which research advances would improve protein sequence analysis.

**Questions:**

None

**Limitations:**

Yes, limitations have been addressed.

**Strengths And Weaknesses:**

Predicting the influence of mutations to proteins is an extremely important problem for both medical and evolutionary genetics. The author's make an important insight here that naively fitting models to available protein sequences, without consideration of the underlying generative process via a phylogeny driven by fitness, is problematic. They demonstrate the problem well, both theoretically and empirically. Finally, they identify compelling paths forward for future research.

---

> ### Author Response · Authors · 2022-08-01
> **Response to Reviewer wgyk**
>
> We thank the reviewer for their thoughtful response.

---

### Official Review · Reviewer_G9Cm · 2022-07-14

**Rating:** 9
**Confidence:** 4
**Soundness:** 4 excellent
**Presentation:** 4 excellent
**Contribution:** 4 excellent

**Summary:**

This paper studies the issue of learning molecular fitness from evolutionary data. The studied data-generating process involves molecules that evolve according to a stochastic mutation process along with a fitness function that effects the survival of the mutations through selection. Due to phylogenetic effects, the true data-generating distribution convolves fitness with evolutionary history, meaning that we cannot estimate fitness just by learning the true data distribution. The authors provide a rigorous explanation of this effect and then go on to explain why despite this issue, empirical approaches that aim to learn the data distribution through MLE or Bayesian inference often can infer fitness quite well. The authors describe two hypotheses, the first of which appears to be the common assumption that previous practitioners had, whereas the second appears to be new hypothesis that the authors propose. Understanding which hypothesis is true has important implications for the field, as hypothesis 2 suggests that better density estimation will not necessarily improve fitness. The authors then construct a diagnostic test to test each hypothesis and validate it on simulated data. The authors then move to experimental data where they show that hypothesis two is predicted by this diagnostic test in the majority of scenarios.

**Questions:**

No questions

**Limitations:**

No issues

**Strengths And Weaknesses:**

Overall, I think this is a great paper. The question studied is important, as estimating fitness from evolutionary data is of great interest in computational biology. The authors explain rigorously the issue of non-identifiability and then go on to provide a novel hypothesis for why existing approaches have empirically been able to estimate fitness nonetheless. The authors show under relatively loose conditions (Thm 4.1) why a mis-specified model may be beneficial for inferring fitness (hypothesis 2) and then go on to show under simulations how to test for hypothesis 2. Finally, they provide strong empirical evidence for hypothesis 2.

For originality, quality, clarity, and significance I would rate this highly.

---

> ### Author Response · Authors · 2022-08-01
> **Response to Reviewer G9Cm**
>
> We thank the reviewer for their thoughtful response.

---

### Official Review · Reviewer_x1wv · 2022-07-21

**Rating:** 7
**Confidence:** 1
**Soundness:** 3 good
**Presentation:** 3 good
**Contribution:** 3 good

**Summary:**

This paper delves into the broad question of why generative sequence models work well in learning from phylogenetic data and points out some directions for improving them. It first starts showing that given simplifying assumptions made by popular models (which try to fit the data distribution $p^0$), it is not possible to infer fitness of a sequence given any amount of data. They suggest that instead, they should model the stationary distribution $p^{\inf}$. Authors utilize these assumptions and show that, in general, fitness is non-identifiable. Thus, they conclude models trying to fit p^0 are misspecified, however, this has some empirical benefits that can be evaluated in a hypothesis testing framework. Their experiments in synthetic data accept the hypothesis that models fitting $p^0$ can indeed get a close approximation to $p^{\inf}$.


**Questions:**

1. One of the main arguments, if not the main one, for theorem 3.2 (non-identifiability) is that the model/prior at hand has to satisfy exchangeability (assumption 3.1). According to authors, in general, exchangeability is hold in models in the literature. Doesn't this suggest that we should attempt to build models that put non-exchangeable priors on the sequence? Also, if we remove this assumption, is the claim in line 194 correct? should it be reworded to "is in general impossible **with current models**"?
2. Can you provide more recent examples that can help to understand how general the exchangeability assumption is?

**Limitations:**

- They recognize that the assumptions made in some parts of the theoretical analysis are not always guaranteed in real practice.

**Strengths And Weaknesses:**

Strengths:
- Draws interesting conclusions on the hardness of learning.
- It proposes future directions that are theoretically more plausible to work.

Weaknesses:
- Although this work has machine learning contributions, I am unsure that NeurIPS is the best venue to review this work. The assumptions that are made in the analysis are very domain specific and it is hard to evaluate if they are sound. For instance, the two exchangeability examples for assumption 3.1 (which is crucial to prove a key theorem of the paper, thm 3.2) date from before 2010. I am unable to tell if the most recent methods also make this assumption, therefore, it is hard to evaluate the generalizability of this assumption.
- The claims made by authors may need to be qualified. See question 1.

---

> ### Author Response · Authors · 2022-08-01
> **Response to Reviewer x1wv**
>
> We thank the reviewer for their thoughtful response. Exchangeability is a fundamental idea in statistics and machine learning, and has less to do with specific model architectures than the dataset itself (https://people.eecs.berkeley.edu/~jordan/courses/260-spring10/lectures/lecture1.pdf). Intuitively, exchangeability says that the order that the data points are observed in doesn't provide any information about the data generating process. To give a commonplace machine learning example: a VAE model applied to MNIST is an exchangeable model, because the probability it assigns to a collection of images doesn't change if we change the order of those images. Exchangeability is a sensible assumption here because we have no information by which to order the images. If, instead of the MNIST dataset, we had a historical database consisting of handwritten digits and the date they were collected (say, 1600, 1750, etc.) then the ordering of the datapoints would matter, and we would want to build a model that used the temporal information. In this case, exchangeability would be violated.
>
> Our Assumption 3.1, then, cannot be avoided simply with better models; it is best understood as an assumption about the kind of data we have. Because sequences are collected at essentially the same time (compared to the timescale of millions of years that we are concerned with) they do not have an informative ordering. Exchangeability would be violated if instead we had (for instance) dated fossils of DNA sequences stretching back millions of years. In this case, we would have informative temporal information, we could build non-exchangeable models, and we could (potentially) identify fitness and phylogeny. However, such data is unavailable.
>
> To our knowledge, all phylogenetic models that are applied to sequences separated by large evolutionary distances satisfy exchangeability (it would be worrisome if they did not). For three recent examples in machine learning venues, see https://proceedings.mlr.press/v161/moretti21a/moretti21a.pdf, https://openreview.net/pdf?id=SJVmjjR9FX, and https://proceedings.neurips.cc/paper/2020/file/d96409bf894217686ba124d7356686c9-Paper.pdf.

---

### Meta-Review · Area_Chair_zspw · 2022-08-22

**Recommendation:** Accept
**Confidence:** Certain

**Metareview:**

This paper presents the argument that molecular fitness is not identifiable from observational sequence data alone and that bigger models trained on bigger datasets will inevitably lead to better fitness estimation. The reviewers found strengths in the paper because "estimating fitness from evolutionary data is of great interest in computational biology", and because they "show under relatively loose conditions (Thm 4.1) why a mis-specified model may be beneficial for inferring fitness (hypothesis 2) and then go on to show under simulations how to test for hypothesis 2". All reviewers highlighted the importance of the problem and the theoretical and empirical contributions. There was a minor concern that this is the best venue for the work and how these results might translate across domains. There is interest across domains in the alignment between the true target of inference (here molecular fitness) and the upstream analysis targets (here density estimation). Overall, the reviewers evaluations were highly favorable and they considered this a valuable contribution to the field.

**Award:**

No

---

### Decision · Program_Chairs · 2022-09-14

Accept